# Quantitative In-Depth Analysis of the Mouse Mast Cell Transcriptome Reveals Organ-Specific Mast Cell Heterogeneity

**DOI:** 10.3390/cells9010211

**Published:** 2020-01-14

**Authors:** Srinivas Akula, Aida Paivandy, Zhirong Fu, Michael Thorpe, Gunnar Pejler, Lars Hellman

**Affiliations:** 1Department of Cell and Molecular Biology, Uppsala University, The Biomedical Center, Box 596, SE-751 24 Uppsala, Sweden; srinivas.akula@icm.uu.se (S.A.); fuzhirong.zju@gmail.com (Z.F.); getmeinahalfpipe@gmail.com (M.T.); 2Department of Medical Biochemistry and Microbiology, BMC, Box 589, SE-751 23 Uppsala, Sweden; aida.paivandy@imbim.uu.se (A.P.); gunnar.pejler@imbim.uu.se (G.P.); 3Department of Anatomy, Physiology and Biochemistry, Swedish University of Agricultural Sciences, SE-75007 Uppsala, Sweden

**Keywords:** transcriptome, mRNA, mast cell, tryptase, chymase, serine protease, FcεRI, heparin, histamine

## Abstract

Mast cells (MCs) are primarily resident hematopoietic tissue cells that are localized at external and internal surfaces of the body where they act in the first line of defense. MCs are found in all studied vertebrates and have also been identified in tunicates, an early chordate. To obtain a detailed insight into the biology of MCs, here we analyzed the transcriptome of MCs from different mouse organs by RNA-seq and PCR-based transcriptomics. We show that MCs at different tissue locations differ substantially in their levels of transcripts coding for the most abundant MC granule proteins, even within the connective tissue type, or mucosal MC niches. We also demonstrate that transcript levels for the major granule proteins, including the various MC-restricted proteases and the heparin core protein, can be several orders of magnitude higher than those coding for various surface receptors and enzymes involved in protease activation, as well as enzymes involved in the synthesis of heparin, histamine, leukotrienes, and prostaglandins. Interestingly, our analyses revealed an almost complete absence in MCs of transcripts coding for cytokines at baseline conditions, indicating that cytokines are primarily produced by activated MCs. Bone marrow-derived MCs (BMMCs) are often used as equivalents of tissue MCs. Here, we show that these cells differ substantially from tissue MCs with regard to their transcriptome. Notably, they showed a transcriptome indicative of relatively immature cells, both with respect to the expression of granule proteases and of various enzymes involved in the processing/synthesis of granule compounds, indicating that care should be taken when extrapolating findings from BMMCs to the in vivo function of tissue-resident MCs. Furthermore, the latter finding indicates that the development of fully mature tissue-resident MCs requires a cytokine milieu beyond what is needed for in vitro differentiation of BMMCs. Altogether, this study provides a comprehensive quantitative view of the transcriptome profile of MCs resident at different tissue locations that builds nicely on previous studies of both the mouse and human transcriptome, and form a solid base for future evolutionary studies of the role of MCs in vertebrate immunity.

## 1. Introduction

Mast cells (MCs) are primarily known for their prominent role in immunoglobulin E (IgE)-mediated allergies [1,2]. However, they are evolutionary old and humans with a complete lack of MCs have not been identified, indicating an important physiological role of these cells. They are found in all studied vertebrates, and MC-like cells have also been identified in tunicates, a chordate, which represents an early evolutionary branch leading to all vertebrates, including mammals [3]. Mast cells most likely act as early sentinel cells, inducing inflammation and attracting other inflammatory cells to the area of inflammation. They are primarily resident hematopoietic tissue cells that are distributed along both external and internal surfaces of the body where they act in the first line of defense [1,2]. Mast cells are frequently found in the connective tissue of the skin and around blood vessels and nerves, as well as in the mucosa of the airways and intestines. Mature MCs are generally not found in the circulation. Two main types of MCs have been identified both in rodents and humans: Connective tissue MCs (CTMCs) and mucosal MCs (MMCs). Connective tissue mast cells are, as the name implies, primarily found in the connective tissue and in the peritoneal cavity whereas MMCs are mainly found in the intestinal mucosa of mice and rats and to a lesser extent in the lungs [4,5,6,7,8,9].

Mature MCs store large numbers of cytoplasmic granules that are rapidly exocytosed following activation. The majority of proteins found in these granules are serine proteases, which can generally be subdivided into chymases and tryptases [10,11,12,13]. Chymases are chymotrypsin-like and cleave substrates after aromatic amino acids whereas tryptases are trypsin-like in their specificity, with a preference for Arg and Lys in the P1 position [10,11,12,13]. Very high amounts of these proteases are found in MCs, where the levels can reach 35% of the total cellular protein [14]. Phylogenetic analyses of the chymases have led to the identification of two distinct subfamilies, the α-chymases and the β-chymases [6,13,15]. Connective tissue mast cells also express an MC-specific carboxypeptidase- carboxypeptidase A3 (Cpa3). These proteases have been shown to inactivate snake, bee, and scorpion toxins; regulate blood pressure by angiotensin II generation; and control inflammation by cleaving a selective panel of cytokines [1,12,16,17,18,19,20]. Mast cell proteases likely have several other important physiological functions, including connective tissue turnover and regulating coagulation [21,22]. A potential role of the mouse MC tryptase (Mcpt6) to link innate and adaptive immunity in the chronic phase of *Trichinella spiral* infection has also been observed [23].

Mast cell granules also contain high levels of heavily sulfated, and thereby negatively charged, glycosaminoglycans, such as heparin or chondroitin sulfate, and also large amounts of vasoactive low molecular weight compounds, including histamine and serotonin [14,24,25,26,27]. Mast cells also express a number of cell surface receptors and other molecules, of which many are MC specific or restricted to a few cell types. Such important receptors are the high affinity receptor for IgE (FcεRI) and the receptors for stem cell factor (SCF) and interleukin -3 (IL-3) [28,29]. Connective tissue mast cells and MMCs show major differences in their expression of granule proteases and also in their levels of receptors and in numerous other aspects, indicating that they have, at least partly, different physiological functions.

To obtain a multi-faceted view of the phenotype of mouse MCs, here we performed a transcriptome analysis of purified mouse peritoneal MCs and analyzed MCs from other tissues for the expression MC-restricted compounds. By using a larger number of cells, as opposed to single cell analysis, we also increased the fidelity of the study to obtain a highly variable estimate of 20% to 30% of the top transcripts and a good quantitative estimate of the transcription levels of approximately all 21,000 mouse genes. Single cell analysis, with 1 to 1.5 million reads per cell, and analysis of the combined information from more than 50 cells may give similar data. However, no such information is available for a more detailed comparison. 

Our results showed that MCs of different tissue locations are highly specialized, differing substantially in their expression of major granule compounds and enzymes involved in the processing of such compounds. This analysis has also resulted in the identification of a number of interesting genes expressed at much higher levels in MCs compared to the other 12 different tissues included in this study. Analyses of the roles of these newly identified proteins in MC biology may also give new insights into the complex physiological roles of this medically important cell. This detailed analysis of the MC transcriptome, and the abundance and heterogeneity of MCs in various mouse organs, can also serve as a solid base for future studies concerning the roles of MCs in vertebrate immunity.

## 2. Materials and Methods

### 2.1. Mice

Female BALB/c mice were purchased from Taconic Biosciences ((Europe) Ejby, Denmark) and maintained at the animal facility in the Biomedical Center (Uppsala University) or the Swedish Veterinary Institute (Uppsala, Sweden). The animal experiments were approved by the local ethical committee (Uppsala djurförsöksetiska nämnd; Dnr 5.8.18-05357/2018).

### 2.2. Generation of Bone Marrow-Derived MCs (BMMCs) and the effect of LPS Stimulation

Bone marrow cells were isolated from the femur and tibia of mice and grown in Dulbecco’s modified Eagle’s medium (Sigma-Aldrich, Saint Louis, MO, USA) containing 30% WEHI-3B-conditoned medium, 10% heat-inactivated fetal bovine serum (BSA) (Gibco, Carlsbad, CA), 100 U/mL penicillin, 100 μg/mL streptomycin, 2 mM L-glutamine (all from Sigma-Aldrich, Saint Louis, MO, USA), and 10 ng/mL IL-3 (PeproTech, Rocky Hill, NJ, USA). The medium was changed twice every week and cells were cultured at a concentration of 0.5 × 10^6^ cells/mL in a humidified 37 °C incubator with 5% CO_2_ for at least 4 weeks to obtain mature and pure BMMCs. The cells were divided into two separate fractions, one was directly frozen in liquid nitrogen for the preparation of total RNA and the second was incubated in the above medium with the addition of 1 μg/mL of LPS per ml for 4 h, after which the cells were pelleted and frozen in liquid nitrogen for subsequent RNA preparation.

### 2.3. Peritoneal Cell Extraction and Sorting of Peritoneal Mast Cells

For the extraction of peritoneal cells, mice were euthanized by isoflurane overdose and neck dislocation, the abdominal skin was removed, and 9 mL of ice-cold phosphate-buffered saline (PBS) was injected into the peritoneal cavity. After shaking the abdomen, peritoneal lavage fluid was collected, and the cells were pelleted by centrifugation at 400× *g* for 10 min. The cells were resuspended in magnetic-activated cell sorting (MACS) buffer, containing 0.5% BSA in PBS pH 7.2, and 2 mM EDTA, followed by incubation with 20 μL of anti c-kit MicroBeads (Miltenyi Biotec, Bergish Gladbach, Germany). After 30 min, cells were washed, resuspended in MACS buffer, and passed through an LS column (Miltenyi Biotec, Bergish Gladbach, Germany). Magnetically labeled (c-kit^+^) cells were collected and used for RNA isolation and assessment of mast cell purity. Alternatively, to enhance the purity of peritoneal MCs, collected peritoneal cells were incubated in the first step with 1 μL of primary PE-Cy5-conjugated anti-lineage antibodies CD3 (17A2), CD4 (GK1.5), CD8b (eBioH35-17.2), CD11b (M1/70), CD19 (ebio1D3), B220 (RA3-6B2), Gr-1 (RB6-8C5), and TER-119 (TER-119). After 20 min, the cells were washed, resuspended in MACS buffer, and incubated with 20 μL Anti-PE-Cy5 MicroBeads (Miltenyi Biotec, Bergish Gladbach, Germany). After 30 min, cells were washed, resuspended in MACS buffer, and passed through an LD column (Miltenyi Biotec, Bergish Gladbach, Germany) according to the manufacturer’s instructions. Subsequently, the unlabeled (Lin^−^) cells were collected, washed, and in the second step incubated with 20 μL of anti c-kit MicroBeads for 30 min. After washing, cells were resuspended in MACS buffer and passed through an LS column (Miltenyi Biotec, Bergish Gladbach, Germany). The unlabeled cells were discarded while magnetically labeled (Lin^−^ c-kit^+^) cells were collected and used for RNA isolation and assessment of mast cell purity. The anti-lineage antibodies were purchased from BD Biosciences (Franklin Lakes, NJ, USA) or eBioscience (Hatfield, United Kingdom). 

### 2.4. Image Analysis

The magnetically sorted single run CD117^+^ cells and the Lin^−^ CD117^+^ cells were cytospun onto glass slides using a Shandon Cytospin 2 (Thermo Fisher Scientific, Inc., Waltham, MA, USA) and were allowed to dry before staining with toluidine blue using a standard protocol to assess the purity of sorted Lin^−^ CD117^+^ MCs. The cells were imaged using a Nikon Eclipse Ni-U microscope (100× or 200× magnifications).

### 2.5. RNA Isolation

RNA isolation from purified cell fractions: Total RNA was prepared from MACS-sorted cells, cultured BMMCs plus 4-h LPS-treated cells using the Nucleospin RNA kit (from Macherey-Nagel, Germany), according to the manufacturer’s recommendations. The RNA was eluted with 30 μL of DEPC-treated water, and the RNA concentration was determined by using a Nanodrop ND-1000 (Nano Drop Technologies, Wilmington, Delaware, USA). The integrity of the RNA was confirmed by visualization on 1.2% agarose gels using ethidium bromide staining.

RNA isolation from tissue: Ear, lungs, brain, tongue, heart, liver, kidney, pancreas, duodenum, proximal part of the colon, spleen, and uterus tissues were dissected from the mouse. Immediately after removal, the tissues were frozen in liquid nitrogen, and made into a fine powder by grinding with a pestle and mortar. The tissue powder was then used for total RNA isolation using the protocol described above.

### 2.6. Analysis of the Transcriptome by RNA-Seq and by the Thermo Fisher Ampliseq Chip and PCR-Based Method 

Total RNA from the different cell fractions and whole tissues were used for transcriptome analyses (GATC-Biotech, Ebersberg, Germany). mRNA was purified by poly-A selection following fragmentation of the RNA and sequencing of 30 million fragments was performed. Individual read lengths of around 50 to 100 nucleotides were then matched against a reference transcriptome library. The FPKM values were normalized based on the total number of reads and sent to us as a large Excel file with all the different cell and tissue samples in one file for easier comparison. The number of reads per gene was, for the RNA-seq data from GATC, adjusted to the transcript length as longer transcripts generate more fragments per mRNA and therefore a higher number of reads. Following the sorting of the 33,915 different transcripts from the most highly expressed to the lowest, we manually went through the entire list to identify the MC-specific highly expressed genes and the genes that were markedly more highly expressed in MCs compared to other tissues. The results from the RNA seq analysis matched well with previous cDNA library screenings and also with the Thermo Fisher chip-based Ampliseq transcriptomic platform (Ion-Torrent next-generation sequencing system- Theromofisher.com). The Thermo Fisher mouse Ampliseq transcriptome analysis platform is based on the purification on a chip of the individual mRNAs (as cDNAs), which are then PCR amplified and sequenced individually. The RNA is not fragmented but copied into cDNAs before binding to the chip. In the Ampliseq analysis, every mRNA is read only once and the number of reads then corresponds to the expression level more directly. Here, as for the RNA seq analysis, all the data from 16 different samples were sent to us as a large Excel file containing all the transcripts for all the 16 samples in one file, which enabled an easy comparison of the transcript levels between cell fractions and tissues. Following the sorting of the 23,931 different transcripts from the most highly expressed to the lowest, we manually went through the entire list to identify the MC-specific highly expressed genes and the genes that were markedly more highly expressed in mast cells compared to other tissues. In total, 265+ such genes were identified for the pure MC preparation and all of these genes are listed in the Appendix A, with the tissue and the expression level in that tissue that showed the second highest expression of this gene being named in the table for an easier comparison.

## 3. Results

### 3.1. Preparation of RNA from Tissues and Purified Peritoneal Cell Fractions

To study the total transcriptome of MCs, we sought to obtain pure populations of cells. One of the best sources of MCs in mice is the peritoneal cavity, where MCs represent ~1% to 2% of the cells. Peritoneal MCs are almost identical in their phenotype to classical skin MCs concerning their major granule protease content, and both of these MC populations are classified as CTMCs. However, they seems to differ in the expression levels of a few other proteins, including Mrgprb8 and 13, AdamtsS1 and 5, Sox3, CD59a, and CD34, which indicate that we cannot take them as direct equivalents [30]. Other cells of the peritoneum are mainly macrophages and B cells, which constitute approximately 60% and 35%, respectively, of the entire peritoneal cell population. Mast cells were purified by positive selection with magnetic cell sorting (MACS) using an anti-c-kit (CD117) reacting monoclonal antibody. Total RNA was prepared from this cell preparation and also from in vitro-differentiated mouse bone marrow-derived MCs (BMMCs) grown in the presence of recombinant IL-3 for 4 weeks, and from total mouse ear and total mouse lungs. These RNA preparations were subjected to transcriptome analyses by two methods: (1) RNA-seq methodology, which involves mRNA purification and fragmentation, cDNA synthesis, and sequencing 30 million reads on these fragments. The reads were then normalized to the length of the individual mRNAs and listed as fractions of the entire transcriptome; and (2) the mouse Ampliseq transcriptome analysis platform, based on the purification of cDNA copies of individual mRNAs on a chip, followed by PCR amplification and individual sequencing. The RNA is not fragmented, and generally every mRNA is therefore read only once, and the number of reads will thereby directly match the expression. Bone marrow-derived mast cells, and the ear and skin samples were of high purity. However, the peritoneal MCs were only 30% to 35% pure, which is why we decided to focus on MC-specific transcripts in this initial analysis of peritoneal MCs (Figure 1A).

### 3.2. Analysis of Transcript Levels in Peritoneal MCs

The data from the transcriptome analysis of the preparation from mouse peritoneal MCs was arranged (Table 1), with a focus on the granule proteins, the enzymes involved in the processing of granule compounds and on MC-specific cell surface receptors. From this preparation, we also made an estimate of the difference in the abundance between house-keeping genes and cell type-specific transcripts. This was done only for the GATC RNA seq, since this method is completely unbiased. The house-keeping genes included different ribosomal proteins, elongation factors, β-actin, and a number of proteins involved in general cell metabolism. When examining the top 100 expressed transcripts in the peritoneal MCs and MCs from the skin and the lungs, approximately 80 of them represented house-keeping genes. A calculation of the transcriptomes of the MCs and the different tissues (skin and lungs), based on the 300 top transcripts, indicated that the house-keeping genes accounted for 60% to 80% of the total mRNA pool of a cell or tissue, with the remaining transcripts being tissue specific (data not shown). From the RNA seq data, it was apparent that the most abundant granule proteins constitute the majority of the cell type-specific transcripts. These included the various granule proteases, which could individually account for several percent of the total MC transcriptome. In the tables, the results from both the RNA seq analysis and the PCR-based (Ampliseq) analysis are presented. This data from the independent approaches matched remarkably well; the total counts vary due to differences in the total number of normalized reads, but it was clear that the relative abundances between individual transcripts were in very good agreement. Two individual samples from the RNA-seq analysis are listed in Table 1 to study the reproducibility of the analysis. As can be seen from the table, the two independent runs give almost identical values, showing that the analysis is stable and reliable. 

The major cell-specific transcripts of the peritoneal MCs were four different proteases, including the serine proteases Mcpt4, Mcpt5, and Mcpt6, and the MC-specific carboxypeptidase Cpa3 (Table 1). Mcpt4 is the major chymotryptic enzyme of CTMCs [6,31]; Mcpt5 is the alpha chymase, which has evolved into an elastolytic enzyme in rodents [32]; and Mcpt6 is the major mouse MC tryptic endopeptidase [33]. In addition to these serine proteases, the peritoneal MCs expressed relatively high levels of Cpa3. All other proteases were expressed at very low levels, almost two orders of magnitude lower than the major protease transcripts (Table 1). Mcpt7, the second MC tryptase, was expressed at a level of 2% compared to Mcpt5 (Table 1). Cathepsin G, a chymotryptic enzyme expressed at high levels in human neutrophils, was expressed at a level of 1.2% compared to Mcpt5. Granzyme B, a serine protease primarily expressed by cytotoxic T cells, was expressed at a level of ~0.4% in comparison with Mcpt5 (Table 1). All other granzymes (A, K, C, D, E, F, and G) were almost undetectable (Table 1). Cathepsin C, the enzyme responsible for N-terminal processing of the majority of the different granule-stored serine proteases, was also expressed at a relatively low level: 0.9% to 2.5% in comparison with Mcpt5, depending on the method used (RNA-seq or Ampliseq) (Table 1). 

One of the most characteristic receptors of MCs is the high affinity receptor for IgE, FcεRI. This receptor has three different subunits that are all encoded by separate genes: The α-, β-, and γ-chains. The α-chain, which is essentially MC and basophil specific, was expressed at approximately 3% or 0.8% of the Mcpt5 levels, respectively, as determined by the two different techniques (Table 1). The receptor for SCF, c-kit, was expressed at approximately half the level of the IgE receptor α-chain (1.5% of Mcpt5 levels) (Table 1). The IL-3 receptor was expressed at even lower levels (~0.5% of the Mcpt5 levels) (Table 1).

When examining a panel of different cytokines, including IL-4, IL-5, IL-6, IL-18. and IL-15, we noted extremely low levels of the corresponding transcripts (Table 1). Out of these cytokines, only IL-15 was expressed at levels above background (0.14% of the Mcpt5 level) (Table 1). 

### 3.3. Analysis of In Vitro Differentiated BMMCs

Cells from the bone marrow of BALB/c mice were grown in the presence of 30% conditioned media from WEHI-3B cells, which contains IL-3, and with a supplement of 10 ng/mL of recombinant IL-3 for 4 weeks. The resulting cell culture consisted of almost 100% pure MC-like cells, i.e., representing BMMCs. Almost all cells stained positively for alcian blue and toluidine blue, although not as strongly as the peritoneal MCs. The BMMCs were analyzed with both RNA seq and Ampliseq. In the BMMCs, high transcript levels were seen for two of the four major protease transcripts, Mcpt5 and Cpa3 (Table 2). In contrast, very low levels of Mcpt4 and also low levels of Mcpt6 were detected (Table 2). Interestingly, a very high level of transcript corresponding to the FcεRI α-chain was seen (Table 2). For this gene, we noted that the RNA-seq and Ampliseq approaches produced different results. With the RNA-seq, the level of the α-chain was more than 50% of the Mcpt5 level, whereas the Ampliseq data indicated ~10%, which was still very high compared to the peritoneal MCs where the levels were approximately 3% and 0.8%, respectively, for the two independent technologies (Table 2).

We also tested the effect of bacterial lipopolysaccharide (LPS) on the transcript levels in BMMCs. After a 4-h incubation with 1 μg/mL LPS, relatively small changes in transcript levels were seen. However, major changes were observed for a few transcripts, including Mcpt4 with a 4- to 5-fold increase, Mcpt7 and Mcpt2 with 4-fold increases, Mcpt1 with a 10- to 12-fold increase, granzyme B with a 4- to 5-fold increase, and a 1000-fold increase for granzyme C (Table 2). However, it should be noted that the baseline levels of granzyme C were very low (Table 2). We also examined the expression of Toll-like receptors (TLRs) in the BMMCs. Using Ampliseq, we only detected significant levels of TLR-4, with 134 and 76 reads for BMMCs and BMMCs-LPS-4 h, respectively, and very low levels of TLR-1, -2, and 6 (six reads or lower).

### 3.4. Analysis of the Transcript Levels in Mouse Ear with a Focus on MC Transcripts

To assess the MC phenotype in various tissues, we performed an analysis of mouse connective tissue and lungs. To assess MCs in a connective tissue environment, ear tissue was selected. To evaluate the phenotype of ear MCs, the levels of MC-specific transcripts in the ear was assessed and was related to major transcripts of the ear. As expected, the most abundant transcripts of the ear were keratins. We detected high levels of at least 12 different keratins in the ear, of which keratin 10, 2, and 14 were the most abundant (Table 3). We also detected relatively high levels of α-actin and troponin C2, revealing the presence of muscular tissue. The different MC-specific transcripts, including proteases, receptors, and processing enzymes, were then used to determine the amount and types of MCs in the ear tissue. The analysis showed that CTMCs were by far the dominating MC subtype in the ear, represented by high levels of transcripts for Mcpt4, Mcpt5, Mcpt6, and Cpa3. In contrast, transcripts for MMC proteases (Mcpt1 and Mcpt2) were essentially undetectable (Table 3), indicating that MMCs were not populating the ear. Interestingly, the levels of Mcpt4 were two-fold higher than Mcpt5, which indicated that the MCs of the ear were more differentiated compared to the peritoneal MCs (Table 3). It was also noted that BMMCs expressed extremely low levels of Mcpt4 in comparison with the ear skin MCs (~0.01% of the Mcpt5 levels) (Table 2). The fact that BMMCs are typically grown in the presence of IL-3 or SCF, or a combination of them, and that cells obtained from such cultures are relatively immature, indicates that additional unknown factors, which are essential for the development of fully mature MCs, are present in both the peritoneum and skin. Two individual samples from the Ampliseq analysis for the ear and the lung are listed in Table 3 and Table 4 to study the reproducibility of the analysis. As can be seen from the tables, the two independent runs give very similar values, showing that the Ampliseq analysis is also stable and reliable.

We also considered the overall proteolytic environment of the skin. We found the expression of a large number of different kallikreins, of which many were expressed at high levels: Primarily kallikrein 7, 5 10, 8, and 11. We also found the expression of several matrix metalloproteases (MMPs), primarily MMP-2, -14, and -3. To control these, there are a number of protease inhibitors, and we found that many of these were expressed at high levels. These included a large panel of serpins (23 members), whereas low levels of SLPI expression were seen (data not shown). Altogether, these analyses showed that a complex network of proteases and protease inhibitors resides in the skin tissue, including significant levels of various MC proteases.

We also found low levels of three defensins, Def-b6 (40 and 28 reads), Def-b1 (90 and 91 reads), and Def-b14 (17 and 23 reads), by the Ampliseq analysis.

### 3.5. Analysis of the Transcript Levels in Mouse Lungs with a Focus on MC Transcripts

To study the number and types of MCs in the lungs, we analyzed the total lung transcriptome. Similar to our assessment of the ear, we used MC granule proteases and MC-specific receptors as primary tools to determine the presence and phenotype of lung MCs. As previously shown by immunohistochemistry, the number of MCs in a normal healthy lung is very low [34,35]. We detected transcripts representative of CTMCs, MMCs, and basophils in the lungs as exemplified by Cpa3, Mcpt5, Mcpt6, and Mcpt4 for CTMCs, Mcpt1, and Mcpt2 for MMCs and Mcpt8 for basophils (Table 4). However, the levels of all of these proteases were extremely low, indicating low numbers of MCs and basophils in the lungs. As expected, the major transcripts of the lungs were for the surfactant-like SFTc, and also very high levels of both lysozyme 1 and 2. High lysozyme levels are most likely important for bacterial clearance of the lungs. We found the presence of transcripts for both mouse lysozyme genes, *Lyz1* and *Lyz2*. However, very low levels of defensin transcripts were observed using the Ampliseq analysis: Only three of these were expressed: Def b2 (11 and 14 normalized reads), Def b1 (1 and 2 normalized reads), and Def b14 (1 and 2 normalized reads), which should be compared to values of over 10,000 reads for Lyz1 (Table 4).

### 3.6. Analysis of Transcript Levels in Peritoneal MCs of a Higher Purity to Obtain a More Complete Picture of the MC Transcriptome

To obtain a more complete picture of the mouse MC transcriptome, we performed several additional attempts to obtain pure peritoneal MC populations, which resulted in the use of a protocol involving several steps of magnetic cell sorting (MACS), resulting in a purity of over 95% as well as a sufficient amount of cells to obtain a good coverage of the entire transcriptome (Figure 1B). Based on cell morphology, the few percent of contaminating cells appeared to be primarily B cells, which have a very different transcriptome from MCs and therefore impose little risk of errors in the identification of transcripts.

Total RNA was purified from pure peritoneal MCs and from 10 additional mouse tissues, both to identify transcripts that were highly expressed in MCs compared to other tissues and to obtain a more complete picture of the presence of MCs in different tissues. The selected tissues were the brain, liver, kidney, duodenum, pancreas, tongue, heart, proximal part of the colon, uterus, and spleen. Based on the previous validation, we used only one method (Ampliseq) to study these tissues.

Similar to the initial analysis of the peritoneal MCs, we decided to focus on the granule proteases (Appendix A) as well as the cell surface receptors (Appendix A).

However, in this new analysis, we also included protease inhibitors (Appendix A), cell adhesion molecules (Appendix A), enzymes involved in heparin, histamine and lipid mediator synthesis (Appendix A), cytokines and chemokines (Appendix A), transcription factors expressed at higher levels in MCs compared to other tissues (Appendix A), RNA binding molecules and splicing factors (Appendix A), molecules involved in cell division (Appendix A), molecules involved in cell signaling and intracellular granule transport (Appendix A), and finally on other category proteins (Appendix A).

The top transcripts were, as in the previous analysis, the granule proteases (Mcpt4, Mcpt5, Mcpt6, and Cpa3; Appendix A). They were expressed more than 100 times higher than most other MC-specific or MC-related transcripts. The only other highly expressed transcripts were the core protein for heparin and chondroitin sulfate (serglycin) (8905 reads) and FcεR-gamma (2913 reads).

The top transcript was Mcpt6, with 67.773 normalized reads (Appendix A). Mcpt5, Mcpt4, and Cpa3 were in the same range, with 45221, 33.040, and 45.604 normalized reads, respectively (Appendix A). Mcpt7 was expressed almost 200 times lower (386 reads) and cathepsin G (1111 reads) approximately 60 times lower than Mcpt6 (Appendix A). Granzyme B was also present but at a level more than 100 times lower than Mcpt6 (581 reads; Appendix A). All the other granzymes as well as the mucosal MC proteases Mcpt1 and 2 were almost undetectable (Appendix A). Tryptase gamma, a membrane-bound tryptase, was also expressed at a very low level (123 reads; Appendix A). We also observed a low level of the basophil-specific protease Mcpt8 (38 reads; Appendix A [8,36]. Interestingly, we detected relatively high levels of some additional proteases not generally considered as MC-related, namely cathepsin E (1248 reads), urokinase (661 reads), tripeptidyl peptidase (TPP) (588 reads), ADAMTS 9 (370 reads) and ubiquitin carboxyl-terminal hydrolase (198 reads) (Appendix A).

The major protease inhibitors found in the MC transcriptome were Serpin b1a, Serpin b6a, Lxn, and Cystatin F (1037, 127, 185, and 181 reads, respectively) (Appendix A).

Of the immunoglobulin receptors, we primarily detected FcεRI alpha (345 reads) (Appendix A) and FcγRIII (752 reads) (Appendix A). The inhibitory receptor FcγRIIb was expressed at a lower level (91 reads) (Appendix A). The signaling component of FcεRI and FcγRIII, i.e., FcεR-gamma, was relatively highly expressed (2913 reads) and the FcεRI beta chain was expressed at somewhat lower levels (1297 reads) (Appendix A). Most of the other Fc receptors were expressed at almost undetectable levels, except the very interesting immunoglobulin-related receptor, MC immunoglobulin-like receptor 1 (MilR1; 283 reads). This receptor, which has been shown to have an important dampening role during MC activation, was expressed at least 100 times higher in MCs than in any of the other tissues analyzed (Appendix A) [37].

Several receptors to cytokines or other mediators were found to be highly expressed in MCs compared to all other tissues. These receptors included the SCF receptor (c-kit; 1198 reads), the IL-33 receptor (ST2; 1693 reads), the IL-3 receptor alpha chain (159 reads), and the common beta chain of the IL-3, IL-5, and GM-CSF receptors (CSF2Rb; 415 reads) (Appendix A).

High expression of the interferon alpha/beta receptor beta chain (Ifnar2) in comparison with other tissues was also seen (453 reads) (Appendix A). A number of other interferon receptors were identified that were also present in other tissues at similar levels, including the alpha chain of the alpha/beta interferon receptor (90 reads) and both the alpha and beta chains of the interferon gamma receptor (228 and 115 reads, respectively) (Appendix A).

Relatively high levels of expression were seen for of a number of the recently discovered receptors: The MAS-related GPR members, including Mrgprb2, Mrgprb1, Mrgpra4, Mrgprb8, Mrgprb4, and Mrgprx1 (899, 366, 155, 19, 11, and 13 reads, respectively) (Appendix A).

Mrgprb2 is known to induce non-IgE receptor-mediated responses to substance P and other small cationic substances [38,39,40].

The endothelin receptor A (Ednra; 892 reads) was also expressed at higher levels in MCs versus all other tissues analyzed (Appendix A).

The beta-adrenergic receptor (Adrb2) was also expressed higher in MCs versus other tissues (631 reads; Appendix A). The ATP receptors P2rx7 and P2rx1 were also highly expressed in MCs compared to other tissues (571 and 327 reads, respectively) (Appendix A). ATP has recently been shown to have potent MC-activating properties through several mechanisms, including the P2rx7 and P2rx1 receptors [41]. CD200r3 was expressed 100- to 1000-fold higher in MCs than in other tissues (275 reads) (Appendix A).

CD200r3 has been shown to be an activating receptor for MCs [42]. High levels of CD34 and CD81 were also seen (332 and 1475 reads, respectively) (Appendix A). We also observed relatively high levels of CD55 (decay accelerating factor or DAF), CD274 (PD-L1), and CD276 (B7-H3) (439, 200, and 163 reads, respectively) (Appendix A). These latter genes were expressed at higher levels in MCs compared to other tissues (Appendix A). Significant levels of the Ig domain-containing and inhibitory receptor CD300a were also detected (184 reads) (Appendix A). We did not detect expression of CD40L and very low levels of CD40 (Appendix A). Very low levels of TLRs were also seen, with TLR-4 being the most abundant (61 reads) (Appendix A), and the only other TLR detected was TLR-13 (13 reads) (Appendix A).

Interestingly, we also detected low levels of expression for a large number of olfactory receptors and also the vomeronasal receptors in MCs but not in any of the other tissues analyzed. However, the expression levels for these were very low, ranging from 18 to 2 reads (Appendix A); the biological relevance of this finding is thus questionable. Interestingly these were also identified as MC expressed in an important study of the MC and basophil transcriptomes by Dwyer et al. [30].

Several other receptors, of which a few have been described as MC-related, were also expressed at comparably high to very high levels compared to other tissues. The most striking of these was a member of the GDNF receptor family, Gfra2 (2016 reads), that binds GDNF and neurturin, Fxyd5/dysadherin (1850 reads) and Lat2 (linker of activation of T-cell family members; 1588 reads) (Appendix A). Lat2 has previously been shown to have a central role in MC activation upon IgE receptor crosslinking [43]. Interestingly, we also observed relatively high levels of Tarm1, the T cell-interacting activating receptor on myeloid cells protein 1 (156 reads; Appendix A).

Tarm1 was much more highly expressed in MCs compared to all other tissues analyzed.

We also examined the different cell adhesion molecules expressed by peritoneal MCs (Appendix A). Some of these were selectively expressed by MCs whereas others were highly expressed in both MCs and other tissues. In MCs, we detected integrins a4, b1, b2, a2b, a9, a5, am, b7, and a2 (404, 444, 388, 266, 217, 135, 113, 211, and 137 reads, respectively) (Appendix A). We also detected relatively high levels of the P selectin ligand CD162 (369 reads) and CD43/sialophorin (134 reads) (Appendix A). Interestingly, we also detected the expression of mucin 13 (147 reads) (Appendix A). For mucin 13, only duodenum and tongue were higher (520 and 1178 reads, respectively) (Appendix A). Very high expression of the GPCR family member of adhesion receptors, Adgre5/CD97 (2839 reads), was also observed (Appendix A).

The next sets of genes to be analyzed were the enzymes and transporters involved in heparin, histamine, and lipid mediator synthesis. As mentioned previously, the most highly expressed of these genes was serglycin, the heparin core protein (8905 reads) (Appendix A). Of the deacetylases involved in sulfation of the carbohydrate backbone of heparin, NDST-2 was essentially MC-specific and quite abundant (1112 reads), whereas the other NDSTs (NDST-1, 3) were almost absent in MCs (Appendix A). A number of other genes that are directly or indirectly involved in heparin, chondroitin sulfate, or heparan sulfate biosynthesis were also expressed at relatively high levels in the MCs compared to most other tissues (Appendix A). The most highly expressed of these were *N*-acetylgalactose amine 4-sulfate 6-*O*-sulfotransferase (Chst15) (448 reads); carbohydrate sulfotransferase 11 (Chst11) (216 reads); idunorate-2-sulfatase (Ids) (657 reads); heparan sulfate 6-*O*-sulfotransferase 2 (Hs6st2) (152 reads); *N*-acetylgalactose aminide alpha-2-6-sialyltransferase 3 (St6galnac3) (290 reads); N-sulfoglucoseamine sulfohydrolase (Sgsh) (215 reads); exostosin 1 (Ext1), the endoplasmic reticulum-resident type II transmembrane glycosyltransferase involved in the chain elongation step in heparan sulfate synthesis (254 reads); and glucosamine-fructose-6-phosphate aminotransferase isomerizing 1 (Gfpt1) (398 reads) (Appendix A).

For histamine synthesis, we found high levels of histidine carboxylase (936 reads) and the monoamine transporter (Slc18a2; transports amines, including histamine) (861 reads) (Appendix A). For serotonin synthesis, we noted high levels of tryptophan hydroxylase (Tph1; 550 reads) and the serotonin transporter (Slc6a4; 1468 reads), which is involved in reabsorption of released serotonin after granule release (Appendix A).

For enzymes involved in lipid mediator synthesis, we saw a number of relatively highly expressed genes, including phospholipase A2 (PLA2 g7; 1116 reads), 5-lipoxygenase (Alox5; 486 reads), prostaglandin D synthase (Hpgds; 450 reads), and lysophosphatidylcholine acetyltransferase 2 (Lpcat2; 177 reads; of importance for Platelet activating factor (PAF) synthesis) (Appendix A).

With respect to the cytokines and chemokines expressed by peritoneal MCs, we noted low or very low levels of most of the classical interleukins, with detectable expression only of IL-4, IL-13, IL-15, IL-18, IL-1α, IL-6, and IL-12b (13, 4, 8, 18, 6, 12, and 9 reads, respectively) (Appendix A). However, we saw substantial levels of two chemokines, Ccl-2 and Ccl-6 (397 and 188 reads, respectively), as well as three cytokines involved in fibroblast growth and differentiation: Fibrosin (Fbrs), TGF-β1, and TGF-β activator (Nrros) (355, 550, and 180 reads, respectively) (Appendix A). We also detected low expression of hepatocyte growth factor (Hgf) and oncostatin M (Osm) (59 and 38 reads, respectively) (Appendix A). Numerous interferons were also found at very low levels, including Ifna9, Ifna2, Ifna4, Ifnb1, Ifna16, Ifna12, Ifna5, Ifne, Ifnab, and Ifna14 (8, 7, 7, 6, 6, 3, 2, 2, 2, and 2 reads, respectively) (not listed in a table).

Several transcription factors were expressed at much higher levels in MCs compared to other tissues. Two of the most well characterized factors for MC development, GATA-2 and MITF, were found among the most highly expressed (2272 and 370 reads, respectively) (Appendix A) [44,45]. In contrast, GATA-1 and GATA-3 were expressed at relatively low levels (74 and 27 reads, respectively) (Appendix A). This is also is in line with previous investigations where GATA-1 is primarily expressed in erythrocytes and platelets, and GATA-3 in NK cells, T cells, and the placenta. A number of other transcription factors were highly expressed in peritoneal MCs compared to other tissues. Among these were Runx1 and Runx3 (638 and 409 reads, respectively) (Appendix A). We also detected extremely high levels of Myb (2490 reads) and very high levels of a number of additional transcription factors, including the stress-induced transcription factor CREB3l1 (992 reads); the zinc finger, E box, and homeobox protein Zeb2 (1019 reads); the transcription factor TOX2 (568 reads); Fli1/ERGB (473 reads); CREBbp, a CREB binding factor (426 reads); Crtc3 (CREB-regulated transcription activator 3; 369 reads); Pnn (pinin transcription factor E box binding (CAGGTG); 423 reads); Ldb1 (LIM domain binding transcription factor; 423 reads); Cbfa2t3 (transcriptional regulator interacting with HDACs and Runx; 418 reads); Cited4 (Cbp/P300 interacting transactivator with Glu/Asp-rich carboxy-terminal domain 4; 277 reads); Atf7ip (activating transcription factor 7-interacting protein 1; 357 reads); Kmt2c (lysine *N*-methyltransferase 2C; 343 reads); Meis2 (a homeobox protein; 300 reads); Ankrd12 (ankyrin repeat domain-containing protein 12; 269 reads); Tal1 (basic helix-loop-helix transcription factor; 246 reads); Samsn1 (Sam domain-containing negative regulator of B cell activation; 187 reads); Pu.1/Spi1 (205 reads); and LAT (111 reads). Interestingly, several of these transcription factors have never been linked to MC development. A number of other transcription factors were also present but at similar levels as in other tissues, including Ikaros (113 reads), and a number if IRFs, including IRF-9, -2, -1, -7, -5, -8, -6, -3, and -4 (101, 100, 81, 65, 60, 40, 37, 25, and 11 reads, respectively) (not listed in the table) [44]. A number of STATs were also found, including STAT-3, -6, 2, -5b, 5a, and -4 (256, 196, 112, 56, and 17 reads, respectively) (not listed in the table) [44]. The STATs were also present in most other tissues. These transcription factors may still be of major importance for the phenotype of the MC, although the majority of them are most likely of lower importance for determining the path during differentiation from the hematopoietic stem cell to become a fully mature MC.

A number of RNA-binding proteins, splicing factors, and proteins involved in cell division were also found to be highly expressed in MCs compared to the other tissues (Appendix A).

As a next step, we focused on proteins involved in cell signaling and vesicular transport. The main such signaling components and vesicular transport proteins are listed in Appendix A. High MC-level proteins found included several phospholipase C isoforms, primarily the B and D types (1001 and 350 reads, respectively), Rgs13 (regulator of G protein signaling; 289 reads; a signaling molecule shown to be involved in IgE-mediated responses), as well as Syk, a central player for IgE receptor signaling [46]. Several of the components known to be involved in IgE receptor signaling, including Lyn, VAV, and Grb2, were also expressed in MCs [46]. Lyn (44 reads) and VAV (204 reads) are also expressed in several other hematopoietic cells at similar levels, and Grb2 (445 reads) was also expressed relatively high in other tissues.

Finally, we noted high expression in MCs of a number of proteins with functions not within the previous categories or with unknown functions. The most striking of these were Lgals (galectin 1), which has immunosuppressive functions (2116 reads), Lrmp (Tap-independent peptide transporter for MHC1; 406 reads), C2 (complement factor 2; 159 reads), and ERV3 (an endogenous full-length retrovirus [47]; 141 reads) (Appendix A).

### 3.7. Analysis of MC-Related Transcript Levels in Mouse Brain, Liver, Kidneys, Tongue, Heart, Pancreas, Duodenum, Proximal Part of the Colon, Total Spleen, and the Uterus.

To obtain a more comprehensive view of the MC transcriptome, we included 10 additional tissues. In these analyses, we observed that the MC transcript levels were very low in many tissues, including the brain, liver, kidneys, colon, pancreas, and heart. In these tissues, the levels of all the CTMC proteases (Mcpt4-6, Cpa3) were below 2.5 reads, and the MMC proteases (Mcpt1 and -2) were undetectable. The levels of the MC proteases were much higher in the duodenum, where the MMC proteases dominated and only low levels of the CTMC proteases were detected (23 and 6 reads for Mcpt1 and 2, respectively). The opposite situation was seen in the tongue, where the CTMC proteases dominated, with only low levels of the MMCs proteases (71, 36, 116, 37, 3, and 0.6 reads for Mcpt4, Mcpt5 and Mcpt6, Cpa3, Mcpt1, and Mcpt2, respectively). Interestingly, relatively high levels of the CTMC proteases were seen in the uterus (14, 8, 25, and 10 reads for Mcpt4, -5, and -6, and Cpa3, respectively), but no transcripts for the MMC proteases could be detected. In the spleen, we noted low levels of both CTMC and MMC proteases (4, 3, 5, 7, 2, and 0.5 reads for Mcpt4, Mcpt5, Mcpt6, Cpa3, Mcpt1, and Mcpt2, respectively).

As a validation of our data, all of the studied tissues showed expression of tissue-specific transcripts, with the intestinal digestive enzymes, chymotrypsinogen, pancreatic lipase, pancreatic carboxypeptidases, trypsinogen, and RNases in the pancreas, albumin as the top transcript in the liver, myoglobin, myosin regulatory chain, myosin alpha chain and cardiac muscle alpha actin in the heart, and myelin basic protein as one of the top transcripts in the brain.

## 4. Discussion

Analyses of total transcriptomes by different techniques offers powerful tools in the analyses of cells, and also allows studies of the development of cells and tissues. For such studies, several alternative techniques have recently been developed. However, remarkably few quantitative studies using such tools have been published. In general, only the fold increase or decrease of transcripts during stimulation is presented and absolute values of expression are rarely given. Data are often also not presented by real numbers but rather as heatmaps, which lack information concerning the abundance of the various transcripts. The presentation of only a fold change can also be misleading concerning their relevance. Frequently, transcript levels can increase 10- or even 100-fold, but when their baseline expression is starting at an extremely low level, it is questionable if such increases are biologically relevant. For example, here we showed that the expression of granzyme C increases by almost 1000-fold after LPS stimulation of BMMCs. However, the original levels were very low (0.1 reads), and we can therefore not be certain that the increase in granzyme C has any biological impact. For this reason, we propose that the absolute values of gene expression (rather than fold changes) are more informative concerning the biological impact. In a hallmark study by Dwyer et al., many MC- and basophil-specific transcripts were identified [30]. Many of the transcripts we describe in this study were also identified in that study. However, it is important to note that the former study by Dwyer et al. did not provide absolute values of expression. Almost all the MC-related transcripts are marked in the same bright red in their heatmaps even if they differ by two to three orders of magnitude in expression levels in our analysis. One of the aims of this study was therefore to generate a deeper understanding of the MC transcriptome by providing data on the absolute expression levels for the various transcripts. This study also involved many more transcripts than the previous study and identified many new proteins that may have major impacts on MC biology, which could be the target for future studies. In contrast to previous studies, this study also involved the analysis of the abundance and heterogeneity of MCs in 12 different mouse organs.

We previously performed a study on rat peritoneal MCs where we measured the absolute values of expression. At that time, the only reliable available method was the use of unamplified cDNA libraries. We therefore hybridized individual filters with a panel of probes able to obtain accurate quantitative information concerning transcript levels for approximately 26 different transcripts (Figure 2) [7].

However, this technique is extremely time consuming and labor intensive, which hampered a more extensive analysis. However, by using the new transcriptome platforms it is now possible to obtain reliable information of almost all the 21,000 genes of the mouse or the human transcriptome, and also information on the abundance of different splice variants. Despite this, these technologies are still relatively poorly validated, mainly because appropriate reference libraries have not yet been established. Our earlier cDNA library data was therefore valuable as a reference material for the optimization and validation process in this study. The possibility of using two independent technologies with the same sample also increased our confidence that the data obtained was reliable. By using the information obtained by both the RNA-seq as well as the chip and PCR-based transcriptome analyses, we can now confirm that the general pattern observed from our previous studies using the unamplified cDNA libraries holds remarkably well.

The transcript levels for the major MC granule proteins are remarkably high, in the range of several percent of the total mRNA pool of the cell [7]. The levels of the most characteristic MC cell surface receptors, including FcεRI, c-kit, the IL-33 receptor, and the IL-3 receptor, were, in general, close to two orders of magnitude lower than for the major granule proteins. The expression of processing enzymes, including cathepsin C (involved in activation of the granule serine proteases), enzymes involved in histamine, serotonin, and heparin synthesis, histidine decarboxylase, tryptophane hydroxylase, and NDST-2, was also relatively low. An estimate of the house-keeping genes indicated that they represent a large fraction of the total transcriptome. A calculation of the transcriptomes of the MCs and the different tissues (skin and lungs), based on the 300 top transcripts, indicated that the house-keeping genes accounted for 60% to 80% of the total mRNA pool of a cell or tissue, with the remaining transcripts being tissue specific. Among the latter, a few major transcripts accounted for the absolute majority, where in the case of the MCs, these transcripts coded for the major granule proteases and the heparin core protein (serglycin).

During MC development, the first proteases to appear are Mcpt5 and Cpa3, whereas Mcpt4 and Mcpt6 appear later and are therefore markers for more mature MCs. An interesting finding in this study was the very low levels of Mcpt4 expression in BMMCs grown in the presence of IL-3. The levels of Mcpt4 were ~1000 times lower than in peritoneal and skin MCs. Mast cells grown in the presence of SCF, a major MC growth factor, have also been shown to express low levels of Mcpt4, indicating that yet unknown factors are needed to obtain full differentiation into mature tissue-resident MCs [45] The level of Mcpt6 was also considerably lower in BMMCs compared to the peritoneal or skin MCs, whereas the IgE receptor α-chain was expressed very highly in BMMCs compared to peritoneal or skin MCs, indicating an immature phenotype of BMMCs. Interestingly, the addition of SCF to the MC cultures seems, at least in C57BL/6 mice, to upregulate Mcpt6 but not to increase the levels of Mcpt4 [45].

When examining the transcriptome of the ear skin and the tongue, we noted that the skin MCs were almost exclusively of the CTMC subtype. The skin and tongue MCs expressed remarkably high levels of four different proteases, of which three were serine proteases and one a carboxypeptidase: Mcpt4, Mcpt5, Mcpt6, and Cpa3 (Table 3). We detected almost no expression of the mucosal MC-specific proteases Mcpt1 and Mcpt2, and no expression of the basophil-specific protease Mcpt8 (Table 3).

When assessing the lungs, we found very low levels of MC- and basophil-related transcripts, confirming the previous findings of such levels in mouse lungs [34,35]. The situation is somewhat different with human lungs, which have relatively high numbers of MCs [48]. This is very relevant for the discussion when using mice as a model for studying the impact of MCs on lung function. However, recently it has been found that MC numbers increase quite substantially in the mouse lung upon different viral infections or allergen challenge [49,50]. Viral infections also lead to increased MC numbers in human lungs [51]. During these infections, it seems that primarily MCs of the mucosal type increase and that this increase is transient, which most likely originates from MC precursors originating from the bone marrow [49,50]. Quite dramatic increases in MC numbers can also appear in other tissues upon infection. Massive increases in mucosal MC numbers are generally observed upon intestinal worm parasite infections [9,52]. An interesting observation comes from *Schistosoma mansoni*-infected BALB/c mice, where a strong upregulation of both MMC and CTMC proteases occurs in the intestinal region, and a very strong upregulation of the basophil-specific transcript for Mcpt8 is seen the lungs, indicating a marked infiltration of basophils [8]. These findings indicate that both MCs and basophils can fluctuate quite dramatically in numbers upon various viral, parasite, and bacterial infections.

By extended analyses of a more than 95% pure MC population from the peritoneum and a simultaneous analysis of 10 additional tissues, we could look deeper into the MC transcriptome and also obtain a more general picture of the presence of MCs and their subtypes in a large panel of different tissues. As for the initial analysis, we noted that the four CTMC proteases are the major tissue-specific transcripts of the CTMC. Next was the core protein for heparin and chondroitin sulfate synthesis, serglycin. When examining the surface receptors, the cell adhesion molecules, and the enzymes involved in histamine, leukotriene, prostaglandin, and PAF synthesis, we noted expression of at least two orders of magnitude lower than for the highest expressed genes. Interestingly, we also saw that FcγRIII was more abundant than the IgE receptor, and that the inhibitory receptor FcγRIIb showed relatively low expression compared to the other two dominating Ig receptors. Also of note, the Ig receptor-related receptor allergin-1 or Milr1 was expressed at a level similar to the alpha chain of the IgE receptor (345 and 283 reads, respectively). This receptor may, together with the inhibitory receptor FcγRIIb, control signaling from the IgE receptor [37]. This analysis has also made it possible to examine in more detail the presence of known and previously unidentified transcription factors, signaling molecules, and molecules involved in vesicular transport. These analyses identified a number of genes that had previously not been linked to MCs, for example, Fxyd5, mucin 13, Myb, Zeb2, Tox2, Runx3, and Fli1. We also found several that have previously been identified as being MC expressed, including Gfra2, histamine receptor 4, and CREB3l1 [30]. Further work will be required to establish the importance of these genes in MC development/function.

Most of the immune-related cytokines, including interleukins and colony-stimulating factor, were expressed at very low levels. They were expressed higher in MCs than in other tissues but were nevertheless very low, indicating that MCs primarily produce cytokines upon activation. The low level of cytokine mRNA thereby confirmed our previous findings from rat peritoneal MCs [7] (Figure 2) and has recently also been confirmed in another study of the effect on chromatin opening by various MC stimuli, where the levels of cytokine mRNA were found to be very low in the absence of stimulatory signals [53]. Interestingly, and in contrast to the interleukins, some chemokines were expressed at relatively high levels in MCs, including Ccl2 and Ccl6. Ccl2, also named monocyte chemoattractant protein 1 (MCP1), may have a role in maintaining sufficient numbers of monocytes/macrophages in the tissue, also during non-inflammatory conditions. Three cytokines of importance for fibroblast growth and differentiation were also found to be relatively highly expressed: TGF-β, fibrosin 1, and TGF-β activator (Nrros), indicating a role of MCs in connective tissue homeostasis. Mast cells may also participate in the normal connective tissue turnover via their proteases. These can be released by piecemeal degranulation during non-inflammatory conditions and can, by degrading fibronectin and activating pro-collagenases, remove damaged connective tissue components and provide space for new such components produced by fibroblasts. The production of fibroblast-activating cytokines by MCs may thus be important for tissue maintenance. In line with this, Mcpt4 knockout mice accumulate excessive connective tissue in an age-dependent manner [31,54]. TGF-β also has a potent anti-inflammatory effect and MCs may thereby have a potent anti-inflammatory role in the tissue, in addition to their pro-inflammatory impact. It is also interesting to note that MCs express high levels of galectin 1, which also has anti-inflammatory properties, and suggests that this expression also adds to the potential anti-inflammatory effects of MCs [55]. The detection of very high levels of the initial enzyme in the steroid hormone synthesis pathways (P450scc) was unexpected. The expression of P450scc in MCs was at least 300 times higher than in any of the other tissues studied. However, low levels of the other enzymes in steroid hormone production were detected, indicating that the first product in this cascade, pregnenolone, may have a physiological role in the connective tissue.

Only very low levels of TLRs were observed. We only detected the expression of two TLRs, TLR-4 and TLR-13, which indicates a low response by normal tissue MCs to TLR ligands during non-inflammatory conditions. Very low levels of CD40 and the CD40 ligand were also observed, indicating that MCs have a minor direct role in B-cell responses, as CD40-CD40L interaction is a key component for such responses. However, cytokines produced by MCs during inflammatory conditions may, if present in sufficient concentrations, have a role in isotype switch regulation. The expression of complement factor 2 (C2) with 159 reads in MCs was also interesting, which was three times higher than in the liver sample with 50 reads. Liver is thought to be the primary site for the production of most complement and coagulation components, and this finding indicates that this is not the situation for all of these factors and that MC actually can be an important production site for C2.

Concerning the role of transcription factors in MC and basophil development, several, including GATA-2 and MITF, are well established as key regulators, and others, like IRF8, STAT5, and C/EBPa, are also central to this process [44,45]. Here, we noted that GATA-2 and MITF are relatively MC specific, whereas IRF8, STAT5, and C/EBPa were more broadly expressed, indicating that some factors may be guiding whereas others are necessary as supporting factors. It will be interesting to determine the role of several of the factors with high or medium MC specificity identified in this transcriptome analysis, for example, Myb, Zeb2, Tox2, Runx3, CREB3l1, and Fli1, for their roles in MC development. Further, a number of factors with a potential role in cell signaling, as well as the control of intracellular calcium signaling and granule release, were identified. Some of these are already well characterized whereas the roles of others in regulating MC function remain to be investigated.

It is apparent from this study that MCs express an extremely complex network of receptors, signaling molecules, and transcription factors. High levels of expression were seen for the activating receptors FcεRI, FcγRIII, and CD200R3, the Mas-related receptors (e.g., Mrgprb2), receptors for ATP, and endothelin receptors. These activating receptors are in turn controlled by a number of negatively acting receptors, including FcγRIIb, CD300a, and Allergin-1, as well as possibly by the soluble factor galectin-1. TGF-β secreted by MCs also has an anti-inflammatory role. Interestingly, the neurotrophin receptor Gfra2 was highly expressed in MCs, and one of its ligands, neuroturin, has been shown to have potent anti-inflammatory effects [56]. Relatively high expression levels were also seen for PD-L1, indicating that MCs also can have a dampening function on cytotoxic T cells (Appendix A). The proteases released by MCs also regulate the activity of endothelin-1 and of a set of Th2-promoting cytokines, including IL-18, IL-33, TSLP, IL-15, and IL-21, by cleavage. Endothelin-1 is inactivated primarily by CPA3, whereas chymase cleaves IL-18, IL-33, and IL-15, and tryptase cleaves TSLP and IL-21 [19,20,57].

Analysis of the different tissues for MC-specific protease transcripts also gave an insight into the presence of MCs in different organs. Very low levels of MC-specific transcripts were seen in many tissues, including the brain, liver, kidney, heart, pancreas, and colon. In contrast, relatively high levels of transcripts corresponding to CTMCs were seen in the ear, tongue, and uterus, and high levels of MMC transcripts were found in the duodenum. We observed relatively large amounts of a panel of keratin transcripts in the tongue, indicating some similarities between the tongue and skin, which may have a role in the preferential presence of CTMCs in these tissues. Interestingly, the keratin genes expressed in the two tissues were quite different, with keratins 10, 2, and 14 dominating in the ear and 13, 4, and 16 dominating in the tongue.

As mentioned already in the beginning of the discussion, a large number of the above described MC-related transcripts were also identified in the hallmark study by Dwyer et al. and in the very important papers from the Immgen consortium [30,58] (www.immgen.org). The ones identified in these studies include Plau, Adamts9, C2, Hpgds, the Mrgpr receptor members, CD200r3, CD34, and histamine receptor 4. In these studies, different MC proteases and IgE receptor components as well as the enzymes involved in histamine and heparin synthesis and transport, the chemokine Ccl2, and also a number of the abovementioned transcription factors and signaling molecules. including GATA-2 and MITF, were identified as being MC expressed. The coherence between these different studies makes us confident that we now have a quite detailed view of the transcriptome of the mouse MC. An important study of the human MC transcriptome has also been published by Motakis et al., which makes it possible to look at similarities and differences between human and mouse MCs [59]. In this study, the transcriptome of human skin MC was analyzed by deep-CAGE sequencing. In MCs isolated from human skin without further culturing, high levels of transcripts for the human MC proteases, including the β-tryptases, Cma1, and CPA3, and interestingly, almost four times higher levels of cathepsin G transcripts than for the second most abundant MC protease, the β-tryptase, were found. This in marked contrast to the mouse, where cathepsin G is expressed at a level of only 1% of the level of the mouse tryptase Mcpt6 (Table 1). They also find very high levels of several of the important cytokine receptors for MC development, including kit and IL1RL1(ST2), and also the α and β chains of the IgE receptor (FcεRI). The levels of kit transcripts in these human MC were actually even higher than the level of transcripts for the β-tryptase [59]. They also found high levels of the human counterpart to the mouse substance P receptor Mrgprb2, the Mrgprx2 [59]. The levels of the different toll-like receptors were also found to be very low in the human skin MCs, as we observed in the mouse [59]. So, although there are clear similarities, there are also major differences between mouse and human MCs as, for example, the transcript levels of the receptors, which appear to be in the same range as the proteases in human MCs whereas they are almost two orders of magnitude lower than the protease transcripts in the mouse.

In conclusion, new methods for transcriptome analysis can now give us high quality, quantitative information concerning the entire transcriptome in various tissues and cell preparations. These methods can serve as powerful tools to study the biology and phenotype of a cell type or an organ. Additionally, the use of tissue-specific expressed genes makes it possible to study individual cell populations within a tissue without the need to purify them from the tissue, which reduces the error in the analysis resulting from difficulties in obtaining quantitative recovery of minor cell populations. Here, these techniques were used to obtain quantitative information concerning MCs in various tissues, high quality data on types and numbers of these cells in different organs, and indications for still unknown tissue factors that are of major importance for MC development. The immature phenotype of BMMCs, which are often used as tissue equivalents in studies of MC biology, also raises questions of how relevant they are for such studies. The detailed analysis of a relatively pure population of peritoneal MCs and its comparison with transcript levels in 10 different mouse tissues also made it possible to identify a large number of new MC-specific proteins, of which many may have a major role in MC biology. However, in-depth analyses of the roles of these newly identified proteins in MC differentiation, inflammatory, anti-inflammatory responses, and in normal tissue homeostasis will be needed before we have the complete picture of their precise role in MC biology. This detailed quantitative analysis of the MC transcriptome in the mouse and the abundance and subtypes of MC in different organs can also serve as a solid base for future studies of the role of MCs in vertebrate immunity. As previously mentioned, MC-like cells have been identified in early chordates, as represented by the sea squirt *Ciona intestialis* (a tunicate) [3]. These cells have been shown to stain positive with toluidine blue, as mammalian MCs, and to contain heparin and histamine, as well as expressing a serine protease with tryptic activity and being able to produce prostaglandin D2 [60]. However, the relationship of this tryptic protease to the human MC tryptase and mouse Mcpt6 or Mcpt7 is not yet established. Zebrafish MC-like cells have also been shown to stain positive with toluidine blue, to express a carboxypeptidase that shows 38% identity with human CPA3 and 64% identity to human pancreatic CPA1, and to stain positive for c-kit [61]. Zebrafish MC-like cells are also dependent on GATA2 and Pu-1 for their development, similarly to mammalian MCs, but do not need GATA-1 [61]. Interestingly, zebrafish MC-like cells also express lysozyme C and MPO, two markers that are not normally associated with mammalian MCs but instead are associated with neutrophils in humans, indicating that there are also clear differences between fish and mammalian MCs [61]. A clearly identifiable homolog to the IgE receptor alpha chain has also been identified in the opossum, a marsupial, with possible similar receptor components in the platypus also [28]. However, it is not yet known if they are expressed in MCs. These studies show that information is also starting to accumulate concerning MCs not only in humans, mice, and rats but also other vertebrates. The recently developed transcriptome platforms can now be used to obtain a more detailed picture concerning the evolution of MC and MC-like cells in different branches of the vertebrate evolutionary tree. This detailed analysis of the mouse MC transcriptome can thereby be used as a solid reference for such studies.

## Figures and Tables

**Figure 1 cells-09-00211-f001:**
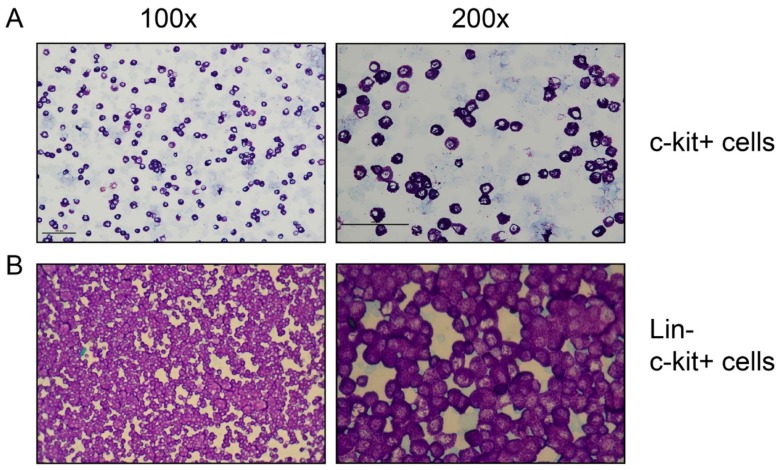
Toluidine blue staining of MACS-separated peritoneal cells. MACS-separated cells were spun onto glass slides by cytospin centrifugation and were stained with toluidine blue. (**A**) the c-kit^+^ cells from the first purification. (**B**) the Lin^−^/ckit^+^ cells from the second, purer, preparation. Both were used for the transcriptome analysis.

**Figure 2 cells-09-00211-f002:**
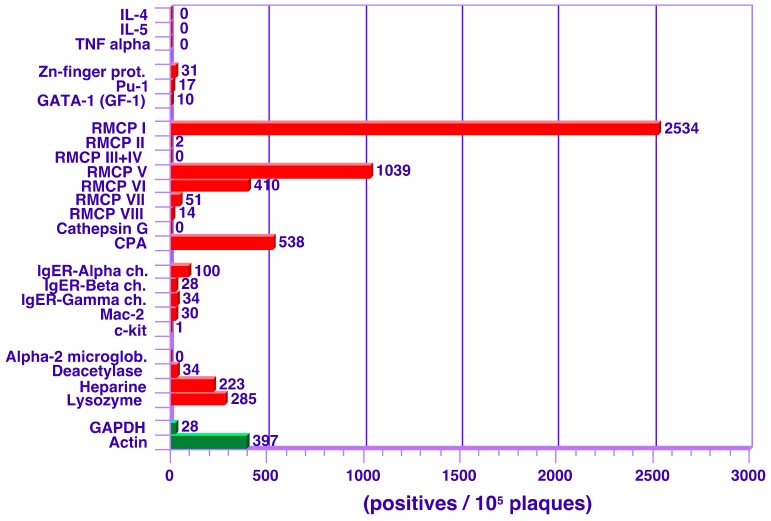
Transcript levels for MC-related genes in rat peritoneal MCs from an earlier study using an unamplified cDNA library [7]. The number of plaques per 100,000 clones were calculated for each of the listed transcripts. These values give an approximate value for the transcript abundance. Approximately 30% of the clones were empty indicating that the relative abundance of for example RMCP-1 was not 2.5% but closer to 3.5% of the total transcriptome.

**Table 1 cells-09-00211-t001:** Transcript levels for granule proteases, cell surface receptors, and enzymes involved in the production and processing of the granule components in mouse peritoneal MCs from BALB/c mice. The number of normalized reads for the proteases is given in actual numbers (obtained from GATC biotech and SciLife Thermo Fisher Ampliseq analyses). Two independent GATC RNA seq runs were performed. For comparison, the relative levels in % to the most highly expressed transcripts (Mcpt5) are indicated in red.

	GATC-RNA Seq		Ampliseq
**Proteases**			
Cma1 (Mcpt5)	17072	17116	35388
Mcpt4 (mMCP-4)	16668	16752	31292
Tpsb2 (Mcpt6)	14244	13548	39628
Cpa3 (CPA-3)	9564	9844	13892
Tpsab1 (Mcpt7)	352	348 (2%)	96
Mcpt9 (mMCP-9)	12	12	0
CtsG (CTS-G)	216	204 (1%)	512
Mcpt8 (mMCP-8)	8	12	12
CtsC (DPP)	148	152 (0.5%)	876
Gzm B	68	72 (0.4%)	236
Gzm A	8	12	12
Gzm K	0	0	2.4
Gzm M	2	4	1.6
Gzm N	0	0	1.2
Gzm C	0	0	0.8
Gzm D	0	0	0.4
Gzm E, F, G	0	0	0
**Receptors**			
FcεRI alpha	480	500 (3%)	252
c-kit	248	252 (1.5%)	720
IL-3R	112	100 (0.5%)	80
**Heparin and Histamine synthesis**			
Srgn (Serglycin)	(920)	(900)(22%)	(2800)
Ndst2	464	564 (2.7%)	1688
Ndst1	68	64	264
Hdc (Histidine decarb.)	300	292 (1.5%)	600
**Cytokines & Chemokines**			
IL-4	12	12	9.2
IL-5	0	0	0
IL-15	12	8	48
IL-18	52	52	0
IL-6	8	8	8

**Table 2 cells-09-00211-t002:** Transcript levels for granule proteases, cell surface receptors, and enzymes involved in the production and processing of the granule components in mouse bone marrow-derived MCs (BMMCs; in vitro differentiated; from BALB/c mice). The number of normalized reads for the proteases is given in actual numbers (obtained from GATC biotech and SciLife Thermo Fisher Ampliseq analyses). Two different growth conditions were tested: unstimulated and stimulated by 1μg/mL of lipopolysaccharide (LPS) for 4 h. The fold increases in expression levels (for Gzm B and Gzm C) after LPS stimulation and the levels (in %) of the FcεRI alpha chain expression compared to Cpa3 are marked in red.

	GATC-RNA Seq		Ampliseq	
	Normal	+4 h LPS	Normal	+4 h LPS
**Proteases**				
Cma1 (Mcpt5)	7102	7618	15683	17413
Mcpt4 (mMCP-4)	5	22	7	34
Tpsb2 (mMCP-6)	746	896	3119	3345
Cpa3 (CPA-3)	13430	11990	22478	17717
Tpsab1 (Mcpt7)	72	241	54	190
Mcpt9 (mMCP-9)	0.9	2.4	0	0
CtsG (CTS-G)	7	9	18	30
Mcpt8 (mMCP-8)	24	25	46	36
Mcpt1 (mMCP-1)	15	64	15	56
Mcpt2 (mMCP-2)	1	10	1.2	15
CtsC (DPP)	51	44	288	243
Gzm B	161	700	386	1900 (4-5x)
Gzm A	0	0	0	0
Gzm K	0	0	0.3	0.9
Gzm M	0.2	0.2	0.3	0.6
Gzm N	0	0	0.1	0.1
Gzm C	0.1	106	0.1	110 (1000x)
Gzm D	0	0	0	0.8
Gzm E	0	0	0.1	0.1
**Receptors**				
FcεRIα	4918	3280	1631	1342 (10–70%)
c-kit	318	307	833	931
IL-3R	30	22	25	22
IL1RL1 (ST2)	1039	1310	4582	5859
CsfRI2B (coβIL-3 GM-CSF)	1851	1855	3485	3499
FcεRIγ	468	432	1815	1545
CD23	0.3	0.5	0	0
FcγRIII	132	135	538	433
FcγRIIb	33	32	40	39
FcγRI	1	1	5	1
CD63	1894	2153	2699	3624
Integrin alpha2b	495	474	2472	2003
**Heparin and Histamine synthesis + Transcr. Factors**				
Srgn (Serglycin)	2637	3139	8107	9990
Ndst2	188	210	820	874
Ndst1	13	7	60	30
5-Lipoxygenase (Alox5)	1029	767	2757	2016
Histidine decarb. (Hdc)	19	99	48	279
Tryp. Hydroxylase (Tph1)	711	1368	2370	4148
Cyp11a1	1840	2041	6422	6252
Slc18a2 (Monoamine transp.)			2218	2348
GATA1	95	81	296	267
GATA2	594	900	5205	7289
GATA3	4	3	14	12

**Table 3 cells-09-00211-t003:** Levels of major transcripts in the ears of BALB/c mice. The number of normalized reads for each of the proteases is given in actual numbers (obtained from GATC biotech and SciLife Thermo Fisher Ampliseq analyses). Two independent Ampliseq runs were performed. Note that the levels of Mcpt4 were very high in the skin compared to the peritoneal MCs.

	GATC-RNA Seq	Ampliseq	
**Major skin transcripts**			
Keratin 10	9490	13724	14386
Keratin 2	6121	15412	18444
Keratin 14	3938	7855	7670
Keratin 5	2726	5304	5637
Keratin 15	1277	2897	2733
Keratin 1	1093	2373	2673
Keratin 77	617	2199	2430
Keratin 79	525	2321	2362
Keratin 17	492	878	780
Keratin 80	171	1453	1574
Loricrin (Lor)	4029	14026	9840
Calmodulin 4	3133	2895	3821
**Mast cell transcripts**			
Mcpt4 (mMCP-4)	110	148	206
Cma1 (Mcpt5)	52	75	79
Tpsb2 (Mcpt6)	38	92	93
Cpa3 (CPA-3)	34	48	42
Mcpt7 (mMCP-7)	8	1.4	1.5
Mcpt2 (mMCP-2)	0.2	0	0
Mcpt1 (mMCP-1)	0.1	0	0
Mcpt8 (mMCP-8)	0	0	0
CtsG (CTS-G)	2	3	6
Srgn (Serglycin)	18	38	48
FcεRIα	0.6	0.1	0.5
FcεRIγ	18	61	56
Ndst2	7	41	45
Ndst1	13	76	83
**Granzymes**			
Gzm C	2	0.95	1.7
Gzm M	3	0.95	1.2
Gzm A	0.5	0.2	0.2
Gzm B	0.15	0.1	0.4
Gzm D	0	0	0

**Table 4 cells-09-00211-t004:** Levels of major transcripts and MC-specific transcripts in lung (BALB/c mice). The number of normalized reads for each of the different proteases is given in actual numbers (obtained from GATC biotech and SciLife Thermo Fisher Ampliseq analyses). Two independent Ampliseq runs were performed.

	GATC-RNA Seq	Ampliseq	
**Major Lung transcripts**			
Scgb1a1 (Uteroglobin)	16	31234	35193
Scgb3a2	1011	1263	1288
Scgb3a1	260	522	522
Scgb1c1	23	39	30
Sftpc (Surfactant)	21382	20059	23191
Sftpa1	1073	9154	9384
Sftpd	434	73	67
Tmsb4x (Thymosin beta 4)	6430	5597	4519
Lyz2 (P-Lysozyme)	4843	10629	10721
Lyz1 (M-Lysozyme)	1131	990	962
Cbr2 (Carbonyl reductase, NADPH2)	2911	6181	4566
Sparc (Osteonectin, BM40)	652	4577	4276
Inmt (Amine-N-methyltransferase)	1947	4363	4730
Sptbn1 (Spectrin beta chain)	280	3742	3432
Epas1 (Endothelial PAS domain cont. prot 1)	642	3710	3439
Cyp2f2 (Cytochrome P450 2F2)	1378	3437	2651
Sec14l3 (Sec 14 like lipid transport)	431	1640	1583
**Mast cell transcripts**			
Cma1 (Mcpt5)	3.7	8	8
Cpa3 (CPA-3)	3.6	1	1
Mcpt4 (mMCP-4)	1	2	3
Tpsb2 (Mcpt6)	0.8	2	2
Mcpt7 (mMCP-7)	0.16	0.1	0
Mcpt8 (mMCP-8)	0.7	3	3
Mcpt1 (mMCP-1)	0.15	0.3	0.4
Mcpt2 (mMCP-2)	0.15	0.4	0.1

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
