# Peer review of "Quantitative In-Depth Analysis of the Mouse Mast Cell Transcriptome Reveals Organ-Specific Mast Cell Heterogeneity"

_cells, 2020, doi:10.3390/cells9010211_

Round 1

Reviewer 1 Report

Please see the uploaded .pdf file

Author Response

Reviewer 1.

The information concerning both methods for transcriptome analysis have now been updated both in the materials and methods and the results sections.

Concerning the GATC RNA-seq analysis the number of reads per sample was 30 million and the FPKM values were normalized at GATC biotech in Germany one of the major RNA seq providers in Europe and the normalizes values were sent to us as a large Excel file where we could sort the values for each tissue according to abundance from highest to lowest expressed genes. The reference library was an GATC inhouse transcriptome reference library. These values of the 33915 transcripts were then manually analyzed and the mast cell related transcripts of major interest were listed in the tables. The analysis was done at the same time for all the cells and tissues of that RNA seq analysis and delivered as normalized reads in a Excel file for easy comparison. Concerning redundancy in Materials and methods section and results. In the results section we only have the information needed to follow the experimental setup whereas in the materials and methods we give much more detail. It is part of the procedure. Concerning table 2 and log fold change we only observed a few transcripts that showed any substantial change in expression level and these are listed in the full level of change.

5-7. The Ampliseq data was generated at the SciLife Lab in Uppsala with the standard 16 sample chip and accompanying reagents from Thermo Fisher. The info and homepage address have been added to the materials and methods section. This relatively new platform is a very convenient platform that gives what we think quite reliable quantitative measurements of the majority of transcripts of a cell listed in a very easy to handle format. Due to the Excel file we could easily compare expression levels of the 23 931 transcripts of the list by just looking manually at transcript after transcript and compare the levels in the pure mast cell population with 10 different mouse tissues. For some of the samples where we had more RNA why we made two fully independent runs just to evaluate the reproducibility of the analysis. This is why we have two rows for the mast cells in Table 1 to evaluate reproducibility of the RNA-seq analysis and two samples of the ear and lung samples in the Ampliseq analysis in Tables 3 and 4 and to evaluate the reproducibility of that analysis in that study. As you can see both methods give very reproducible results as the two independent runs listed in these tables are almost identical. It took me almost two months full time to go through all the 23 931 transcripts manually and to extract the 265+ transcripts of major interest that now is listed in the supplementary tables and in detail are described and discussed in Results and Discussion sections.

I know that it is very much information and we have tried to make tables which is more accurate and quantitative than heat maps due to that the actual values are listed in tables. We had such tables for all the data in tables 1-4 the first version of the manuscript. However, when re-reading the manuscript we realized that these tables just added a lot of space but did not add anything to the understanding. This primarily due to that the four major proteases are 100 times more abundant than anything else why there were four high blocks and then all other genes were almost undetectable. The same would be for a heat Map. Four transcript would be bright red the remaining 260 would be dark blue. By giving the exact reads it is possible to in detail evaluate difference in levels between transcripts that are 10 time, 100 times or 1000 times higher or lower in transcript level with high accuracy and also to see the difference between, for example, transcription factors that are expressed at a level of 15, 155 or 345 reads with high accuracy and evaluate their potential relative importance. Such differences would be impossible from a heat map where all of these transcripts would be dark blue and it would not be possible to distinguish any difference between them. For people in the field I think the separation into different tables and parts of the results and Discussion focusing on different types of proteins make it relatively easy to follow and understand the importance of these transcripts. However, for people outside of the field it may seem overwhelming with the 265+ different transcripts. The readers outside of the field will probably focus on the info that is presented in the relatively simple tables 1-4 and the section on the levels of mast cell related transcripts in the various tissues.

Concerning data from single cell analysis, it is possible that the information from carefully selected single cell analysis of a relatively homogenous population of as said by the reviewer when using high number of reads and more than 50 cells that information can be obtain of the same depth and accuracy as obtained here but I have not seen any such attempts in our field to be able to compare. (A section concerning this issue has been added to the Intro as suggested by the reviewer (Marked in red). However, it would be very interesting to have access to such data. Single cell analysis is ground breaking in the analysis of lineage commitment and there the use of heatmaps are a good tool to show the difference in expression of the top 10-top 20 transcripts specifying the particular cell fate. However, for this analysis performed in this manuscript where we look at differences spanning over more than 3 orders of magnitude (almost 4) and of more than 260 selected transcripts I see little added value by adding a heatmap. A heatmap can possibly at best visualize a difference in abundance of one order of magnitude most likely a good representation only of a factor five in difference and here we have almost 4 orders of magnitude difference between low and high level expression. One such example of more than 3 orders of magnitude is between the proteases and histamine receptor 4. So I agree that heatmaps and diagrams can be good tools in many cases but I do not think they add much value to this particular study as the difference in expression levels are so large.

Reviewer 2 Report

A well written manuscript describing in details gene expression patterns in pure mast cell populations and comparing with expression patterns in while mouse tissues. The manuscript also gives interesting information on the phenotype and abundance of mast cells present in a number of normal mouse tissues.

A few questions to be addressed:

The labels next to Fig 1A and in the legend of Fig 1 are different regarding what we see in Fig 1A. Which one of the two is the right description? The nomenclature of proteases in Lines 262-270 and in Table 2 are not similar. Labels appearing on the text do not appear on the Table. Also there are some discrepancies in nomenclature of proteases between the different tables (i.e. Tpsb2 (mMCP-6) in Table 2, but Tpsb2(Mcpt6) in Table 3). Eliminating these discrepancies would be useful, especially for readers with limited expertise with mast cell proteases. An idea that is repeated a. number of times in the analysis of transcripts from highly purified peritoneal mast cells and the comparison with mouse whole tissues is “.. a number of other transcription factors were also present but at similar levels as in other tissues …” (lines 488-489). What are the authors try to say with this statement? I would suspect that when we find similar levels of a certain transcript with transcripts in whole tissue, this transcript could still have a role in mast cell biology, even though does not differentiate mast cells from other cells. Lines 512-514: “To obtain a more comprehensive view of the MC transcriptome we included ten additional In these analyses we observed that MC numbers were very low in many tissues, includingthe brain, liver, kidneys, colon, pancreas and heart.” The authors state “mast cell numbers”, but this depend on the transcripts they identify. Is this an accurate interpretation of the data? In the Discussion section this is discussed, but maybe the statement could be softened in the Results section.

Author Response

Reviewer 2.

The error in the figure legend has now been corrected. Thanks for observing.

The discrepancy between text and tables concerning the use of mMCP- or Mcpt has now been corrected. It is just different nomenclature for the same proteins but it is good to be consistent. Thanks also for observing this discrepancy.

Concerning the transcription factors. It is fully correct as the reviewer states that transcription factors expressed at the same levels in several tissues may be of major importance for the cell however they are seldom the ones essential for that a cell takes that particular path in a process of differentiation. A section clarifying this has been added to the results section (marked in red).

The statement concerning mast cell number has now been softened in the results section by saying transcript levels instead of MC numbers (marked in red).

Round 2

Reviewer 1 Report

Dear Authors,

Thank you for the detailed response to my requests. In overall, I find that your responses answer adequately my comments.

I was intrigued by the sentence in the main manuscript that “Data are often also not presented by real numbers but rather as heatmaps, which lack information concerning the abundance of the various transcripts” (lines 781-782). However, the last sentence of your response clarifies that it’s this particular study (and possibly other similar studies) where heatmaps and diagrams may not be of value. I believe that actual heatmaps or other diagrams of your choice (the ones that represent the data in the best possible way) could possibly convince me even more about this.

In any case, I believe that the manuscript is well written and conveys important information to the mast cell community. For this reason, I recommend it to be accepted for publication in its present form.

Author Response

As no extra issue were requested by the reviewer I just include the response to the Editor.

Dear Dr Teodora Banu,

We have now tried to address all the very relevant issues brought up by you in your letter.

All the changes to the manuscript have been marked in red.

We have now added a greater comparison to the papers you suggest at several positions in the manuscript.

A large section on the findings in the Motakis paper on human mast cells have been added to the last parts of the discussion. There we discuss the low levels of the TLRs in that article and the difference in expression levels between proteases and receptors seen between these two studies on mouse and human mast cells.

We have also added a large section on the findings related to this manuscript of the data presented in both the Dwyer article and the papers from the Immgen consortium.

Both of these other studies support our findings as you so correctly state in your letter.

We have also added a section on the supportive information on the cytokine expression from the Cildir article as you suggested.

We have now modified the text in the end of the abstract and instead say ¨ Altogether, this study provides a comprehensive quantitative view of the transcriptome profile of MCs resident at different tissue locations, that builds nicely on previous studies of both the mouse and human transcriptome, and form a solid base for future evolutionary studies of the role of MCs in vertebrate immunity.¨

Concerning the peritoneal mast cells and their similarity to skin mast cells we have now modified the text to also add the information concerning this issue from the Dwyer article as follows

Peritoneal MCs are almost identical in their phenotype to classical skin MCs concerning their major granule protease content, and both of these MC populations are classified as CTMCs. However, they seems to differ in the expression levels of a few other proteins including Mrgprb8 and 13 , AdamtsS1 and 5 , Sox3, CD59a and CD34, which indicate that we cannot take them as direct equivalents (30).

Finally as mentioned above we have inserted a large section on the comparison of our data with the information concerning proteases and receptor levels in human mast cells  from the Motakis article.

Thanks for the very good comments which I think really has resulted in a marked improvement of the manuscript.

We hope that these changes now make the manuscript suitable for publishing in this special issue of Cells.

Sincerely

Lars Hellman PhD Professor

Uppsala University

Dept. of Cell and Molecular Biology

BMC, Box 596

SE-751 24 Uppsala

SWEDEN